# MicroRNA-365a/b-3p as a Potential Biomarker for Hypertrophic Scars

**DOI:** 10.3390/ijms23116117

**Published:** 2022-05-30

**Authors:** Joon Seok Lee, Gyeong Hwa Kim, Jong Ho Lee, Jeong Yeop Ryu, Eun Jung Oh, Hyun Mi Kim, Suin Kwak, Keun Hur, Ho Yun Chung

**Affiliations:** 1Department of Plastic and Reconstructive Surgery, School of Medicine, Kyungpook National University, Daegu 41944, Korea; leejspo@knu.ac.kr (J.S.L.); clerk0823@naver.com (J.H.L.); rjyflying@naver.com (J.Y.R.); fullrest74@hanmail.net (E.J.O.); sarang7939@naver.com (H.M.K.); 2Department of Biochemistry and Cell Biology, School of Medicine, Kyungpook National University, Daegu 41199, Korea; med.aurora1106@gmail.com; 3CMRI, School of Medicine, Kyungpook National University, Daegu 41944, Korea; 4BK21 FOUR KNU Convergence Educational Program of Biomedical Science for Creative Future Talents, Department of Biomedical Science, School of Medicine, Kyungpook National University, Daegu 41199, Korea; suin8349@naver.com; 5Kyungpook National University Bio-Medical Research Institute, Kyungpook National University, Daegu 41944, Korea

**Keywords:** microRNA, hypertrophic scar, myofibroblast

## Abstract

The clinical aspects of hypertrophic scarring vary according to personal constitution and body part. However, the mechanism of hypertrophic scar (HS) formation remains unclear. MicroRNAs (miRNAs) are known to contribute to HS formation, however, their detailed role remains unknown. In this study, candidate miRNAs were identified and analyzed as biomarkers of hypertrophic scarring for future clinical applications. HSfibroblasts and normal skin fibroblasts from patients were used for profiling and validation of miRNAs. An HS mouse model with xenografted human skin on nude mice was established. The miRNA expression between normal human, normal mouse, and mouse HS skin tissues was compared. Circulating miRNA expression levels in the serum of normal mice and mice with HSs were also analyzed. Ten upregulated and twenty-one downregulated miRNAs were detected. Among these, miR-365a/b-3p and miR-16-5p were identified as candidate miRNAs with statistically significant differences; miR-365a/b-3p was significantly upregulated (*p* = 0.0244). In mouse studies, miR-365a/b-3p expression levels in skin tissue and serum were higher in mice with HSs than in the control group. These results indicate that miRNAs contribute to hypertrophic scarring and that miR-365a/b-3p may be considered a potential biomarker for HS formation.

## 1. Introduction

Scarring, caused by trauma, refers to the change in the appearance and histopathology of normal skin in diverse clinical aspects and the degree of mitigation. Hypertrophic scars (HSs) are pathological scars induced by surgery, burn injuries, or trauma during the healing process [1] and vary with body constitution. HSs occur most commonly in the outer layers of the skin and arthroses, resulting in damage to individual appearance and severe tissue dysfunction, including itchiness, susceptibility to infection, and pain [2,3]. HSs are well established as a type of tissue fibrosis caused by extracellular matrix (ECM) accumulation and exhibit a robust inflammatory response and fibroblast proliferation [4]. Collagen type I is the main structural element of ECM. It plays a critical role in the development and progression of HSs, and its expression is increased in HS tissues [2,5]. Therapeutic strategies for HS include surgery, radiotherapy, and combination therapy; however, the efficacy of these treatments has not yet been established.

In recent years, although scar formation has attracted significant attention with respect to experimental and genomic factors, its exact mechanisms underlying regulation and recurrence remain unclear. It is generally accepted that biochemical stimulation during wound healing, involving physical tension/strain, transforming growth factor-beta 1 gene expression, and abnormal proliferation of fibroblasts is one of the main causes of pathological scarring. Management of scarring by the analysis of genes and their regulating factors has also been investigated. Recently, the molecular mechanisms underlying the pathogenesis of hypertrophic scarring have drawn much attention, providing potential for the use of gene therapy to treat HSs. Increasing evidence indicates that microRNAs (miRNAs), including those of various genes that regulate ECM deposition and fibroblast hyperplasia, are involved in the progression of hypertrophic scarring.

miRNAs are a class of non-coding RNAs with an average length of 20 nucleotides that play important roles in protein translation, cutting, or deforming target mRNA. Most miRNAs are transcribed from DNA sequences into primary miRNAs, processed into precursor miRNAs, and finally into mature miRNAs. In this study, we analyzed the effect of miRNAs on HS compared with normal skin. Increasing evidence indicates that miRNAs play an important role in cancer and are closely associated with tumorigenesis and prognosis. MiRNAs also have putative roles in myofibroblast regulation, and thus contribute to hypertrophic scarring of the skin. Previous studies have shown that miRNAs may regulate proteins, such as collagen type I, that are known to play a role in myofibroblast regulation and function [6]. As multiple miRNAs have been reported to be aberrantly expressed during HS formation, potential miRNA candidates for HS therapy should be investigated.

In the present study, we analyzed the effect of miRNAs on HS compared with that on normal skin. Furthermore, candidate miRNAs were identified and analyzed as biomarkers of hypertrophic scarring, as circulating mRNAs may be applied for predicting hypertrophic scar formation during healing from operation or trauma and be considered as potential biomarkers for the treatment of HSs based on their expression level in further studies.

In the present study, we used the sample for profiling and validation to select the microRNA and verified the microRNA with specific high expressions in the hypertrophic tissue and serum through the xenograft model, and candidate miRNAs were identified and analyzed as biomarkers of hypertrophic scarring in future clinical applications.

## 2. Results

### 2.1. MicroRNA Profiling and Validation in Groups A and B

#### 2.1.1. Comparative Analyses of microRNAs in HSs

For comparative analyses, we selected miRNAs tightly associated with HS formation, which exhibited more than a 1.2-fold difference between groups A and B. Among these, 10 miRNAs exhibited at least a 1.2-fold increase in expression and 21 miRNAs showed a decrease in expression in group B compared with those in group A (Table 1). The miRNA with significantly different expression were then selected. The relative expression levels of hs-miR-365a/b-3p was significantly increased (*p* = 0.0348) whereas that of hs-miR-16-5p was decreased (*p* = 0.0065) in group B compared to group A (Table 1). All miRNAs were differentially expressed by at least two standard deviations above the background (using the nCounter miRNA array system) in descending order.

#### 2.1.2. MicroRNA-365a/b-3p Is Highly Expressed in HSs

To validate the expression of miR-365a/b-3p and miR-16-5p in groups A and B, we used TaqMan microRNA assays. The expression of miR-365a/b-3p and miR-16-5p was compared with a selection of profiling miRNAs in group A and B. Whereas miR-365a/b-3p expression showed a significant increase in group B, miR-16-5p showed a decrease in expression in group B; however, the difference was not statistically significant (Figure 1).

### 2.2. Hypertrophic Scar Mouse Model

#### 2.2.1. Morphologic Observations

The transplanted human skin grafts remained clinically viable throughout the course of the experiment. In two animals, the graft was lost to mechanical shear in the early postoperative period. The wound healed with contraction and without any complications. The remaining 18 animals grafted with human full-thickness xenografts developed red, thickened scars consistent with human HSs. These scars were firm on palpation and inelastic compared with the surrounding normal skin. Following the full-thickness skin grafting, the observations performed at four, eight and 16 weeks revealed that HSs were formed at eight weeks postoperatively and were markedly visible. These mice represented group E.

#### 2.2.2. Histological Analysis

Histopathological features of human HSs were also observed in proliferative xenograft scars but were noticeably absent in both nude mice and in normal human skin. Histologically, accumulation of dermal collagen was observed using trichrome staining, and in immunohistochemistry using human COL1A1/COL3A1 antibodies in groups C and D. Staining confirmed the engraftment of full-thickness human skin on nude mice at all time intervals, indicating HS formation on human skin. Furthermore, the component ratio of human origin COL1A1 in xenografted nude mouse paraffin sections tended to increase at eight weeks, showing stable full-thickness engraftment and scar formation (Figure 2).

### 2.3. Comparison of Tissue miRNA Expression in Groups C, D, and E

The expression level of miR-365a/b-3p extracted from the scar tissue of group D was significantly higher than that of groups C and E (*p* = 0.0245), validating the function of miR-365a/b-3p. Similarly, a relatively higher level of hs-miR-365a/b-3p was observed in group E than in group C. Contrary to quantitative miRNA analysis, miR-16-5p levels were increased in scar tissues, although these differences were not statistically significant (Figure 3).

### 2.4. Comparison of Serum Circulating miRNA Expression between Groups F and G

As miR-365a/b-3p was overexpressed in group E, the following experiment was performed to measure the effect of xenografted scars on the level of circulating miRNA-365a/b-3p. After skin grafting, nude mice were monitored until postoperative week 16, and serum miR-365 levels were determined using a NanoPhotometer N60 spectrophotometer. The value of circulating miR-365a/b-3p extracted from the serum was significantly higher in group F than in group G (*p* = 0.0354). This result was in agreement with the tissue miRNA expression analysis results in human HSs. However, miR-16-5p expression was significantly higher in group G than in group F (*p* = 0.0281), contrasting the miRNA expression levels in tissue samples (Figure 4).

## 3. Discussion

Surgery for trauma or oncological treatment is an invasive procedure that causes skin injury. Normal and pathological wound healing processes are complex [7,8,9,10]. Histologically, an HS is characterized by excessive fibroblast and mast cell proliferation, accompanied with excessive ECM accumulation [11]. When a wound develops, it is reported to initiate the proliferation phase, where fibroblasts differentiate into myofibroblasts, and collagen is synthesized to accelerate wound healing [10]. If myofibroblasts somehow skip apoptosis after participating in the initial tissue contraction, HS formation occurs [12,13]. Previous studies have suggested an association between abnormal miRNA expression and progression of skin fibrosis [14,15,16]. Thus, analyzing target genes that regulate fibroblast formation and their associated miRNAs may help untangle the underlying mechanism and contribute to the use of miRNAs in the treatment of HSs.

A previous study confirmed that miR-365 is downregulated in colon cancer tissues, is involved in tumor progression, and regulates cancer cell behavior by targeting cyclin D1 and Bcl-2 [17]. miR-365 is also overexpressed both in cells and cutaneous squamous cell carcinoma tissues, suggesting that miR-365 may act as an oncogene [18]. This miRNA also serves as a tumor suppressor gene in gastric cancer [19]. In skin squamous cell carcinoma, miR-365a-3p plays an oncogenic role by downregulating nuclear factor IB to promote CDK6 and CDK4 expression, leading to Rb phosphorylation and tumor progression [12,13,18]. Recent studies have indicated that miR-365 was upregulated in breast cancer [12] Additionally, high expression levels of miR-365a-3p were detected in breast and pancreatic cancer, and are known to play a role in promoting tumor development [20,21]. Moreover, miR-365a-3p affects cell apoptosis, mitosis, migration, and invasion, and ti has been shown to promote laryngeal squamous cell carcinoma xenograft tumor growth and metastases in the liver and hepatic lymph nodes in mouse models, as well as to promote lung carcinogenesis via downregulation of the USP33/SLIT2/ROBO1 signaling pathway [22]. However, to the best of our knowledge, there are no studies reporting the expression levels and functions of miR-365 in HS. Overall, the functions of miR-365a-3p in cancer are complex.

Collagen is an ECM component, and its disorganized accumulation can result in scar formation [23]. Changes in collagen have been reported to play a key role in HS formation, as deposition of collagens I and III can result in HSs [24]. Some studies have shown that excessive scar fibrosis occurs when expression of ECM proteins (such as pro-COL1A1) increase [25]. The idea to induce HSs on nude mice using human skin-xenograft was derived from previous studies that demonstrated equivalent intrinsic properties between xenograft nude mouse-derived scars and human HSs [26,27]. Immunohistochemical staining showed that the expression of COL1A1 in xenografted HS tissue is markedly higher than that in normal tissues, which agrees with recent findings [28]. Another study identified miR-365 to be mechano-responsive, deeply related with end-plate chondrocyte degeneration [29,30], and associated with cardiac hypertrophy [31,32]. In these studies, miR-365 significantly stimulated chondrocyte proliferation and differentiation by increasing the expression of the Indian hedgehog gene and hypertrophic marker type X collagen, and functioned as a positive regulator of cardiac hypertrophy, leading to an increase in the size of cardiomyocytes. Although these studies were limited to stretch-stress-related cardiac cells, bone cells, and chondrocytes, they elucidated the connection between mechanical tension, shearing force, and hypertrophic scarring. Furthermore, in the present study, the role of miRNA-365 in scar formation was reinforced with increased formation in HS tissue, xenografted scar tissue in nude mice, and circulating levels in a human scar xenografted nude mouse model.

Investigation of the miRNA expression profile is gaining popularity because they are key regulators in gene expression networks, influence many biological processes, and might be used as disease biomarkers. Several miRNA gene therapies have been attempted as new therapeutic approaches for treating fibrotic disorders. In particular, antagonizing endogenously upregulated miRNAs using antisense strands has been proposed [33]. Although miRNAs have several roles in fibrosis and have attracted attention as new targets for gene therapy, the regulation and function of miRNAs in hypertrophic scar tissue remain unknown. Some studies have suggested a potential association between miRNAs and HS; for example, the expression levels of miR-21 were reported to be increased in HS-derived fibroblasts, and inhibiting miR-21 expression significantly slowed the formation of HS in vivo [34]. Conversely, the expression levels of miR-137 were markedly decreased in HS, which induced the proliferation and metastasis of fibroblasts [35].

In the present study, we performed a comprehensive analysis of miRNA expression in human HS myofibroblasts and normal skin fibroblasts, and then explored the role of miR-365a/b-3p. The measurement of circulating miRNAs in human HS xenografted nude mice showed the highest fold-change in miR-365a/b-3p compared with the control group (Figure 1). The expression of miR-365a/b-3p was also detected both in normal and HS tissues, which confirmed the significantly increased miR-365a/b-3p expression and decreased miR-16-5p expression in HS tissues. Additional experiments performed on in vivo scar-induced xenograft mouse models for further evaluation of miRNA-365a/b-3p [27] suggested that HS tissue was formed by COL1A1 expression, resulting from the overexpression of miR-365a/b-3p (Figure 3).

Circulating miRNAs isolated from the serum of normal skin and human skin-xenografted nude mice showed significant differences, whereas xenograft nude mice showed significantly increased levels of miRNA-365a/b-3p and miRNA-16-5p, and nude mice with normal skin showed no expression of these miRNAs (Figure 4). These findings indicate that circulating miRNA-365a/b-3p originates from xenograft-derived HSs and could thus be a valuable biomarker for the prediction and risk estimation of HSs. However, miRNA-16-5p level was significantly decreased in tissues, but its circulating levels were increased both with and without statistical significance. This lack of consistency implies that additional studies are needed to elucidate the role of miRNA-16-5p as a regulator of scar formation. In addition, miRNA measurement in the tissue could involve homology for each species, so expression was measured, but in the case of the circulating microRNA measured in the serum, homology does not show, so it mostly did not show any expression (Figure 2 and Figure 4). Overall, our study suggests that miRNA-365a/b-3p is a mediator for regulating hypertrophic scarring in both tissues and serum, and it may be a potential therapeutic target and predictor of hypertrophic scarring through preoperative blood tests.

In conclusion, the present study reports the following major findings. First, miRNA-365a/b-3p was overexpressed in HS tissues. Second, miR-365a/b-3p expression induces hypertrophic scarring by targeting COL1A1, and human skin xenografted nude mice showed a marked increase in circulating miR-365a/b-3p serum levels, suggesting that the scar derived from human skin facilitated the upregulation of circulating miR-365a/b-3p. miR-365a/b-3p levels increased both in tissue profiles and in serum, suggesting that miR-365a/b-3b is a promising biomarker for preventing and treating HS formation.

## 4. Materials and Methods

This prospective single-center study was approved by the Institutional Review Board (IRB) of Kyungpook National University Hospital (Approval No.:2017-04-029-001). The animal experiment was approved by the Institutional Animal Care and Use Committee of Kyungpook National University (Approval No.: KNU 2020-0048) and conducted according to their recommendations. Seven groups were established for the experiments: Group A: fibroblasts from human normal skin, Group B: myofibroblasts from human HSs, Group C: normal skin tissue from human participants, Group D: normal skin tissue from mice, Group E: scar tissue from HS mice, Group F: serum from normal mice, Group G: serum from HS mice (Table 2).

Groups A and B were used for in vitro miRNA profiling and validation, respectively. After establishing an HS mouse model for the in vivo study, comparative analysis of miRNA expression in tissues was performed between groups C, D, and E. Serum circulating miRNA levels were measured in groups E and G (Figure 5).

### 4.1. Human Fibroblasts and Myofibroblasts Culture

Six normal skin tissues and six HS tissues were obtained from patients in the plastic and reconstructive surgery departments. Before surgery, all patients were informed of the purpose and procedure of this study, and the patients agreed to donate excess tissue. Written informed consent was obtained from all participants or their legal representatives. The study was conducted in accordance with the principles of the Declaration of Helsinki.

Normal skin and HS tissues were washed several times with phosphate-buffered saline (pH 7.4) and incubated with 5 mL of Dispase II (Gibco/BRL, Gaithersburg, MD, USA) for 12 h. The dermis and epidermis were separated using forceps. The separated dermal tissues were cut into small pieces and incubated with 0.2% collagenase type II (Worthington Biochemical Corp. Lakewood, NJ. USA) for 1 h and the cells were precipitated by centrifugation at 1000 rpm for 5 min. After several washes, the cell suspension was filtered through a 70 µm cell strainer (SPL Life Science, Gyeonggi-do, Korea). The filtered cells were grown in Dulbecco’s modified Eagle’s medium (Gibco/BRL) containing 100 units/mL penicillin G, 100 g/mL streptomycin sulfate (Sigma, Aldrich, St. Louis, MO, USA), and 10% fetal bovine serum (Gibco/BR) in 100 mm culture dishes in an incubator at 37° and 5% CO_2_. Once the fibroblasts and myofibroblasts reached 70–80% confluence, they were treated with Trypsin 0.25% (1×) solution (Hyclone Laboratories Inc., Logan, UT, USA) for separation.

### 4.2. MicroRNA Profiling and Validation

The miRNAs were extracted from human-derived cells (groups A and B) using QIAzol lysis reagent (Qiagen, Valencia, CA, USA) according to the manufacturer’s instructions. The quantity and condition of the RNA were determined and quality-controlled using an AATI Fragment Analyzer (Denovix Inc., Wilmington, DE, USA) (Figure 5).

The miRNAs from 12 human-derived cells were analyzed for specific biomarkers using the NanoString nCounter Human miRNA Panel version 3a (NanoString Technologies, Inc., Seattle, WA, USA) according to the manufacturer’s protocol. Data analysis was performed using the nSolver™ software (NanoString Technologies, Inc., to obtain the miRNA expression in each sample as a fold-change value; a fold-change of ± 1.2 was considered to be significant.

The relative expression of miRNAs was validated using TaqMan microRNA assays (Applied Biosystems, Foster City, CA, USA). The 2^−^^ΔΔCt^ method was used to determine the relative gene expression. The miRNA expression levels were normalized to RNU6B, which was used as an endogenous control.

A paired Student’s *t*-test and analysis of variance (ANOVA) were used to analyze the miRNA expression data. Statistical significance was considered at *p* < 0.05. Statistical analyses were performed using GraphPad Prism version 7.0 (GraphPad Software, San Diego, CA, USA).

### 4.3. Hypertrophic Scar Mouse Model

#### 4.3.1. Transplantation of Skin Xenografts

Twenty nude mice (athymic NCr-nu/nu, 6 week-old, male, Hana, Busan, Korea) were used in this study. The animals were conditioned for two weeks. All animals were placed under specific pathogen-free biohazard-free conditions. Excess full-thickness normal human skin was obtained from patients during skin graft surgery, and excessive subcutaneous fat was removed. Full-thickness skin was cut into approximately 2.0 × 2.0 cm grafts and placed in sterile saline. Mice were anesthetized using isoflurane (Halocarbon Laboratories, River Edge, NJ, USA). The surgical sites were disinfected using iodine ethanol and then wiped with 70% ethanol. Full-thickness skin (1.5 × 2 cm) was excised from the backs of mice, and normal human full-thickness skin of the same size was transplanted onto the excision area. The margins were sutured with 5-0 Ethilon followed by Bactigras^TM^ (Bactigras©, chlorhexidine acetate 0.05% plus paraffin; Smith & Nephew, Watford, UK) sheet application, saline wet dressing, and subsequent tie-over dressing using silk 3-0. Gross and histologic findings were evaluated at four, eight, and 16 weeks after the operation. This experimental group was designated group E (Figure 5).

#### 4.3.2. Histological Analysis

Tissue samples from xenograft nude mice were excised and fixed in 4% formalin until use. Xenografted nude mouse tissues were excised at the start of the experiment and four, eight, and 16 weeks after skin grafting. All samples were embedded in paraffin, sectioned at 4-μm-thickness, and mounted on slides. The slides were deparaffinized and stained with standard hematoxylin & eosin (H & E) or Masson’s trichrome. Immunohistochemical staining for collagen 1 alpha 1 (COL1A1; 1:1500; Abcam, Wales, UK) and collagen 3 alpha 1 (1:200, Novus Biologicals, Littleton, CO, USA) was performed as previously described using the DAKO kit (Dako Corp., Carpentaria, CA, USA). All slides were analyzed under a light microscope (Leica Microsystems, Wetlar, Germany).

### 4.4. Comparison of microRNA Expression

#### 4.4.1. Tissue miRNAs

Total RNA was extracted from groups C, D, and E for comparative analysis using the QIAzol lysis reagent (Qiagen) following the manufacturer’s instructions. The RNA samples were checked for quantity and quality on a NanoPhotometer N60 spectrophotometer (Implen NanoPhotometer, Westlake Village, CA, USA) and then stored at −80 °C until use (Figure 5).

#### 4.4.2. Serum Circulating miRNAs

Circulating miRNAs were isolated from serum samples obtained from groups F and G using the miRNeasy serum/plasma micro kit (Qiagen). RNA concentration and quality were measured and verified using a NanoPhotometer N60 spectrophotometer (Implen NanoPhotometer). For normalizing circulating miRNA expression, synthetic *Caenorhabditis elegans* miRNA-39 (*cel-miR-39*, Applied Biosystems) was added to the serum samples, as described previously [36] (Figure 5).

#### 4.4.3. miRNA Expression Analysis

The relative miRNA expression was validated using TaqMan miRNA assays (Applied Biosystems). The 2^−^^ΔΔCt^ method was used to determine relative gene expression. The miRNA expression levels were normalized to RNU6B, which was used as an endogenous control. The relative expression levels of circulating miRNAs were normalized to *cel-miR-39*, which was used as a spike-in-control.

#### 4.4.4. Statistical Analysis

Data are presented as the mean ± standard deviation of three experimental replicates. ANOVA was performed using SPSS version 22.0 (SPSS Chicago, IL, USA) to analyze the differences in miRNA expression levels between groups C, D, and E and in circulating miRNA expression levels between groups D and E. Statistical significance was set at *p* < 0.05.

## Figures and Tables

**Figure 1 ijms-23-06117-f001:**
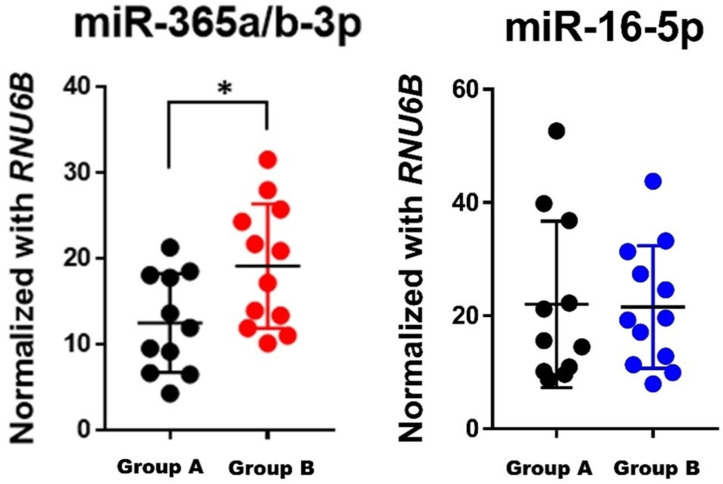
Validation of miRNA in groups A and B. The relative expression of miR-365a/b-3p was significantly higher in group B than in group A (*p* = 0.0244) (* *p* < 0.05). Contrarily, miR-16-5p relative expression was lower in group B than in group A. mRNA expression was validated using TaqMan microRNA assays (Applied Biosystems).

**Figure 2 ijms-23-06117-f002:**
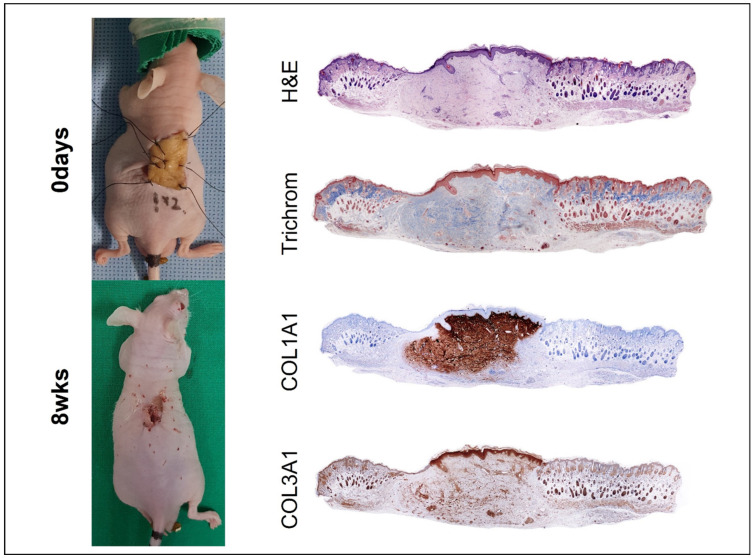
Mouse experimental model of hypertrophic scar formation. All samples in group E were embedded in paraffin, sectioned at 4-μm-thickness, and mounted on slides. The slides were deparaffinized and stained with standard H & E or Masson’s trichrome. Immunohistochemical staining of Collagen 1 alpha 1 (COL1A1; 1:1500; Abcam) and Collagen 3 alpha 1 (COL3A1; 1:200; Novus Biologicals) was performed as described using a DAKO kit (Dako Corp.). All slides were analyzed using a light microscope (Leica Microsystems).

**Figure 3 ijms-23-06117-f003:**
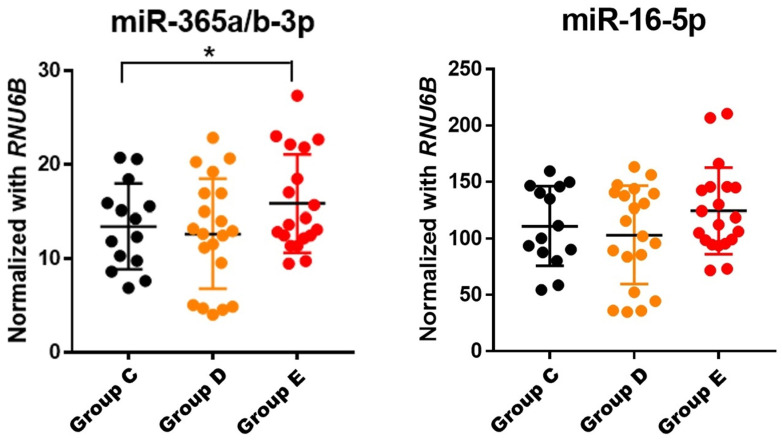
Comparison of microRNA expression levels in tissues: the expression level of miR-365a/b-3p in group C, D, and E. The expression of miR-365a/b-3p was significantly higher in group E compared with that in group C (*p* = 0.0245). There was no statistically significant difference in miR-16-5p levels among the three groups (* *p <* 0.05). Prior to expression analysis, the RNA quantity and quality were verified on a NanoPhotometer N60 spectrophotometer (Implen NanoPhotometer) and then stored at −80 °C until use.

**Figure 4 ijms-23-06117-f004:**
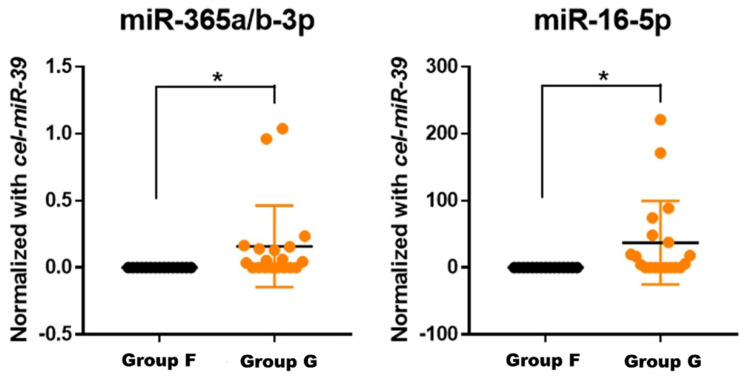
Comparison of microRNA expression in serum samples. The relative expression of miRNAs was validated using TaqMan microRNA assays (Applied Biosystems). Group G showed significantly increased circulating miR-365a/b-3p levels compared with group F (*p* = 0.0354); the miR-16-5p expression level in group G also increased significantly (*p* = 0.0281) compared with that in group F (* *p* < 0.05).

**Figure 5 ijms-23-06117-f005:**
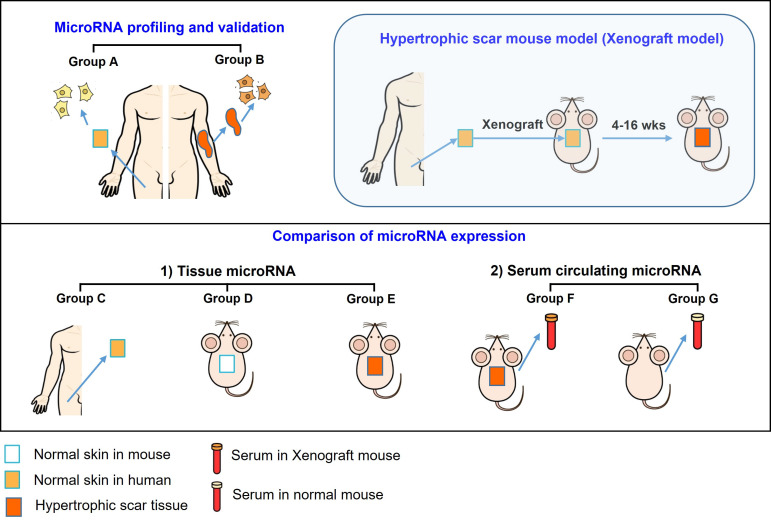
Schematic diagram of the experiments.

**Table 1 ijms-23-06117-t001:** Differential expression profiles of miRNAs in groups A and B. The miRNAs from 12 human-derived cells were analyzed for use as specific biomarkers via the NanoString nCounter Human miRNA Panel version 3a (NanoString Technologies, Seattle, WA, USA) according to the manufacturer’s protocol. NanoString data were used to obtain the miRNA expression in each sample as a fold-change value, and a fold-change of ±1.2 was considered significant using paired *t*-test. *p* < 0.05 was considered to be statistically significant.

**A.** Upregulated microRNA.		
**miRNA**	**Fold-Change** **(Group B/Group A)**	***p*-Value ^a^**
hsa-miR-365a-3p + hsa-miR-365b-3p	1.33	0.0348
hsa-miR-379-5p	1.32	0.2359
hsa-miR-543	1.32	0.2844
hsa-miR-4488	1.31	0.3533
hsa-miR-34a-5p	1.28	0.1264
hsa-miR-299-5p	1.24	0.2089
hsa-miR-99b-5p	1.24	0.2771
hsa-miR-100-5p	1.24	0.3068
hsa-miR-130a-5p	1.24	0.3220
hsa-miR-495-3p	1.20	0.2609
**B.** Downregulated microRNA.		
**miRNA**	**Fold-Change** **(Group B/Group A)**	***p*-Value ^a^**
hsa-miR-424-5p	−1.80	0.2053
hsa-miR-4454 + hsa-miR-7975	−1.75	0.2150
hsa-miR-16-5p	−1.72	0.0065
hsa-miR-21-5p	−1.69	0.2125
hsa-miR-145-5p	−1.67	0.1443
hsa-let-7d-5p	−1.63	0.2070
hsa-let-7c-5p	−1.48	0.1572
hsa-let-7i-5p	−1.46	0.1137
hsa-miR-29b-3p	−1.45	0.3413
hsa-let-7b-5p	−1.44	0.1809
hsa-let-7g-5p	−1.40	0.1952
hsa-miR-503-5p	−1.38	0.2442
hsa-miR-22-3p	−1.37	0.1242
hsa-miR-23a-3p	−1.32	0.3370
hsa-miR-26a-5p	−1.31	0.2804
hsa-miR-29a-3p	−1.28	0.3848
hsa-let-7e-5p	−1.27	0.4229
hsa-miR-221-3p	−1.26	0.2858
hsa-miR-25-3p	−1.23	0.0765
hsa-miR-125b-5p	−1.21	0.3549
hsa-miR-199a-3p + hsa-miR-199b-3p	−1.21	0.5158

^a^ Group A vs Group B, *t*-test, paired.

**Table 2 ijms-23-06117-t002:** Groups of experimental study.

	Group	Components
MicroRNA profiling and validation	A	fibroblasts from human normal skin
	B	myofibroblasts from human HSs
Tissue microRNA	C	normal skin tissue from human participants
	D	normal skin tissue from mice
	E	scar tissue from hypertrophic scar mice
Serum circulating microRNA	F	serum from normal mice
	G	serum from hypertrophic scar mice

## Data Availability

The data presented in this study are available in the manuscript.

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
