# Peer review of "MicroRNA-365a/b-3p as a Potential Biomarker for Hypertrophic Scars"

_ijms, 2022, doi:10.3390/ijms23116117_

Round 1
Reviewer 1 Report
The paper « MicroRNA-365a/b-3p as a Potential Biomarker for Hyper- 2 trophic Scars » written by Joon Seok Lee, Gyeonghwa Kim2 ,Jong Ho Lee, Jung Yeop Ryu, Eun Jung Oh, Hyun Mi Kim, Suin Kwak, Keun Hur, and Ho Yun Chung is written nicely but has some errors throughout that must be corrected to assure comprehension of the data.
Materials and Methods.
Why was human serum data not added too as a control profile. Mouse serum was so it would be necessary to have human also.
Results
Section 3.3. It seems that the Groups should be D, E and F and not C, D, and E. OR why did you compare human to mice in this results section. Please confirm and re-write the section as it is even mentioned that E is higher than C but nothing with E and D and there should be another bar to show the significance.
Section 3.4 In this section again it is not well explained as G to F is higher but there should be an explanation that there was no detection at all for the comparison and that it is only slightly higher. This could be better organized and presented.
Figure 4. Are they really all the same level in Group F and negative detection as this makes the data difficult to assess from the way the figure is presented. There is little explanation in the text and this really should be elaborated.
Therefore, it is necessary to correct the labelling of the Groups and what was compared and assure throughout the manuscript that these changes are in line.
Author Response
Materials and Methods.
Why was human serum data not added too as a control profile. Mouse serum was so it would be necessary to have human also.
- Thank you for your thoughtful comment. We made an effort to divide the groups and heighten the level of understanding. In this paper, we used xenograft and verified the human origin skin grafting in the nude mouse, and the microRNA was measured in the serum after confirming that the scar is human origin, so that was why we did not include human serum.
Results
Section 3.3. It seems that the Groups should be D, E and F and not C, D, and E. OR why did you compare human to mice in this results section. Please confirm and re-write the section as it is even mentioned that E is higher than C but nothing with E and D and there should be another bar to show the significance.
- We apologize for the insufficient explanation. The C, D, E that we listed are the correct expressions, and in this section, the tissue was used to confirm the validation of the microRNA. The confirmation of RNA expression in human tissue was also added as an aspect to be identified. The expression of human tissue and normal mouse was measured similarly, and it was confirmed that it is expressed more in the hypertrophic scar. This aspect was verified by former research through the xenograft experiment, and since it is human origin hypertrophic scar, the confirmation of RNA expression in human hypertrophic scar tissue seems unnecessary. Thank you for your helpful comment.
Section 3.4 In this section again it is not well explained as G to F is higher but there should be an explanation that there was no detection at all for the comparison and that it is only slightly higher. This could be better organized and presented.
- In Figure 1, miR-365a/b-3p increased, and miR-16-5p decreased although there was no statistical significance. Based on this result, we made progress with the assumption that there will be similar progress in the serum. As a result, miR-365a/b-3p showed the same significance as Figure 1 with an increase, and miR-16-5p also showed significance with an increase so it is difficult to interpret that miR-16-5p is related to hypertrophic scars. Figure 1 was also added to paragraph 3.4. We apologize for the insufficient explanation.
- As we explained in line 365~367 in the Discussion section, it seems difficult to understand this miR-16-5p as the marker. Thank you.
Figure 4. Are they really all the same level in Group F and negative detection as this makes the data difficult to assess from the way the figure is presented. There is little explanation in the text and this really should be elaborated.
Therefore, it is necessary to correct the labelling of the Groups and what was compared and assure throughout the manuscript that these changes are in line.
We apologize for the lack of explanation. First, in the case of the microRNA that was implemented with the tissue, there is homology even with the different species (human, mouse) so it can be detected and that shows in Figure 3. Figure 4 defects the circulating microRNA in the serum so the homology cannot appear so it shows a nearly zero value. The explanation was added in lines 375-378 in the Discussion section. Once again, thank you very much for your detailed review about our paper.
Reviewer 2 Report
This study topic is interesting. Overall, this report has good quality and the authors have provided some results to support the significance of this study. Reasonable revisions are needed before acceptance.
Comments and suggestions:
1, the reason for choosing/designing this study need to be explained more
2, an illustration figure about this study is suggested
3, more figures/tables/results are needed for this report
4, more background and refs about scar repair are suggested to be cited/discussed, such as: Chemical Engineering Journal,2022, 134690;Chinese Chemical Letters,2020, 31 (6), 1612-1615
5, the language need to be double checked
Author Response
Comments and suggestions:
1, the reason for choosing/designing this study need to be explained more
- We apologize for the lack of explanation. Our research is a biomarker-based research where we analyzed microRNA that eventually induces hypertrophic scars and measured it in the serum to warn the possible risks of hypertrophic scars. We consider that our research can also be applied in treatment. We have added it in the last part of the Introduction section. Thank you for your comment.
2, an illustration figure about this study is suggested
- Yes, we have created a Figure 1 and have added it.
3, more figures/tables/results are needed for this report
- We added the study protocol as you previously mentioned, and we also added a table about the experiment groups. Our paper has become more resolute all thanks to your comment.
4, more background and refs about scar repair are suggested to be cited/discussed, such as: Chemical Engineering Journal,2022, 134690;Chinese Chemical Letters,2020, 31 (6), 1612-1615
- It was added to the Reference and cited. Thank you for introducing a wonderful paper.
5, the language need to be double checked
We had our paper corrected by a professional English proofreading service again. Thank you.